# Influence of Manufacturing Process in Structural Health Monitoring and Mechanical Behaviour of CNT Reinforced CFRP and Ti6Al4V Multi-Material Joints

**DOI:** 10.3390/polym13152488

**Published:** 2021-07-28

**Authors:** S. Dasilva, A. Jimenez-Suarez, E. Rodríguez, S. G. Prolongo

**Affiliations:** 1Materials Science and Engineering Area, Escuela Superior de Ciencias Experimentales y Tecnología, Universidad Rey Juan Carlos, Calle Tulipán s/n, Móstoles, 28933 Madrid, Spain; s.dasilva.2017@alumnos.urjc.es (S.D.); alberto.jimenez.suarez@urjc.es (A.J.-S); 2Advanced Materials Department, Aimen Technology Center, Polígono Industrial de Cataboi, SUR-PPI-2 (Sector 2), Parcela 3, O Porriño, 36418 Pontevedra, Spain; erodriguez@aimen.es

**Keywords:** multi-material joint, structural health monitoring, multiscale CFRP, Ti6Al4V

## Abstract

Co-cured multi-material metal–polymer composites joints are recent interesting structural materials for locally reinforcing a structure in specific areas of high structural requirements, in fibre metal laminates and lightweight high-performance structures. The influence of manufacturing processes on the morphological quality and their mechanical behaviour has been analysed on joints constituted by sol-gel treated Ti6Al4V and carbon fibre reinforced composites (CFRP). In addition, carbon nanotubes (CNT) have been added to an epoxy matrix to develop multiscale CNT reinforced CFRP, increasing their electrical conductivity and allowing their structural health monitoring (SHM). Mechanical behaviour of manufactured multi-material joints is analysed by the measurement of lap shear strength (LSS) and Mode I adhesive fracture energy (GIC) using double cantilever beam specimens (DCB). It has been proven that the addition of MWCNT improves the conductivity of the multi-material joints, even including surface treatment with sol-gel, allowing structural health monitoring (SHM). Moreover, it has been proven that the manufacturing process affects the polymer interface thickness and the porosity, which strongly influence the mechanical and SHM behaviour. On the one hand, the increase in the adhesive layer thickness leads to a great improvement in mode I fracture energy. On the other hand, a lower interface thickness enhances the SHM sensibility due to the proximity between MWCNT and layers of conductive substrates, carbon woven and titanium alloy.

## 1. Introduction

The addition of carbon nanotubes as reinforcements for carbon fibre reinforced composites (multiscale CFRP) has led to multiple advantages. The improvement of mechanical properties such as interlaminar shear strength, fracture toughness, etc. [1,2,3,4]; and electrical properties, allowing structural health monitoring (SHM), especially via the thickness direction [5], could be the most remarkable. The addition of CNT has been widely studied in combination with structural adhesives in dissimilar metal-composite joints, allowing the structural health monitoring of the adhesive joint [6].

Different methods have been used for the incorporation of nano-reinforcements on composite materials: from spray application on prepregs [7,8], the use of nanotube veils or bucky papers [9,10], to the growth of nanotubes by chemical vapor deposition on the substrate [11]. However, manufacturing multiscale composites using nano-reinforced resins, whether as prepreg, by hand lay-up (HLU), or by resin infusion (VARIM) process, remains one of the easiest, cheapest, and most scalable solutions on a large scale. Furthermore, this method can be used in the manufacturing of multi-materials joints between metallic parts and multiscale CFRP.

The use of metal-composite multi-material structures manufactured by the co-curing process can be an interesting option to locally reinforce a structure in specific areas of high structural requirements, in fibre metal laminates and lightweight high-performance structures [12,13,14]. Special emphasis must be placed on the surface preparation of the metal part, since, as several previous studies have revealed [15,16], it plays a fundamental role in the quality of the joint and, therefore, of the multi-material. The combination of mechanical surface treatment, like grit blasting, with a chemical bonding, such as chemical etching or a sol-gel coating, has shown excellent behaviour [17]. In fact, previously obtained results confirmed that CNT addition to multi-material joints improves the fracture resistance of the joints and allows their structural health monitoring (SHM).

Despite the advantages mentioned above, the use of nano-reinforced resins for multiscale structure manufacturing, especially regarding resin infusion processes, could present material heterogeneities due to the filtering effect. This fact, in addition to influencing the mechanical properties, could cause diversities in electrical conductivity through the material. Furthermore, previous studies [18] have shown that the addition of CNT to epoxy resin increases its viscosity, hindering the composite manufacture and also reduces the Tg of the cured polymer matrix. This must be considered in infusion processes where resin viscosity and its kinetics are especially important for material quality and successful manufacturing.

When the quality of an adhesive joint is analysed, several parameters could influence its final performance, such as the porosity and interface thickness. The influence of porosity either in adhesive joints or in CFRP materials is well known, as it would act as a crack initiator and decrease their mechanical performance. 

Several studies [19,20,21,22] have been carried out regarding the influence of the adhesive interface thickness on their mechanical properties under different stresses. Gleich et al. [20] affirmed that a low adhesive layer thickness results in the highest static joint strength. In fact, they confirmed that a lower thickness on single lap joints promoted lower peel stresses. On the other hand, the influence of adhesive layer thickness on the fracture resistance, fracture energy in mode I, has been studied by several authors [21,22] who have concluded that it increases monotonically with the adhesive layer thickness until reaching a maximum value. So far, however, there has been little discussion on the influence of quality parameters such as porosity and interface thickness, resulting from different manufacturing processes, in co-cured adhesive multi-material joints with CNT doped resin.

The focus of our work is the study of the mechanical performance and SHM ability of multi-material based on titanium alloy, Ti6Al4V, and multiscale CFRP with CNT doped resins. Different processes were applied to manufacture these multi-material joints: hand lay up (HLU) assisted by vacuum bag and vacuum-assisted resin infusion moulding (VARIM). The manufactured materials can be used in fiber-metal laminates (FML) or lightweight high-performance structures.

## 2. Materials and Methods

### 2.1. Materials

The multi-material joints were constituted by two dissimilar substrates: titanium alloy and multiscale CFRP. Ti6Al4V or grade 5 titanium plate with 2 mm of thickness, under AMS 4911N requirements, was selected for this study. Multiscale CFRP was manufactured with a carbon fibre textile reinforcement HexForce G0933 A 1500 TCT 3K 5H SATIN supplied by Hexcel and an epoxy resin doped with multiwalled carbon nanotubes (MWCNT). The polymer matrix is a hot curing epoxy system formed by epoxy monomer, Araldite LY556, in combination with the amine hardener, XB 3473, both supplied by Huntsman. This resin was doped with MWCNT NC7000, produced by catalytic chemical vapour deposition by Nanocyl, with an average diameter of 9.5 nm and 1.5 µm of average length.

### 2.2. Manufacturing

CNTs were dispersed into the resin by the calendering method. Calendering is a mechanical dispersion method, where the resin with CNT passes a certain number of times between the rollers of the calender. This method was based on the use of a three-roll calender machine (Exakt 120S). In each step, the distance between rollers is reduced, so that gradually the agglomerates of CNT are dissolved. The gap size varied in steps (in microns): (1) 120–40, (2) 60–20, (3) 45–15, and (4) 15–5. The last step was repeated four consecutive times. The speed used was 250 rpm.

These parameters and the quality of the dispersion have been studied by the authors in previous works [23,24,25]. A concentration of 0.1 wt% CNT was dispersed in epoxy monomer, since Thostenson and Chou [26] found that the electrical percolation threshold was below 0.1 wt% for carbon nanotubes dispersed in epoxy monomer by calendaring approach. In the present study, the same concentration of CNT (0.1 wt%) and the same dispersion method were used for the fabrication of the joints using the two different manufacturing processes: Hand lay-up (HLU) and vacuum-assisted resin infusion moulding (VARIM). 

The multi-material joints were manufactured directly over the metal part. The set-up was similar in both cases: HLU with vacuum bag consolidation and VARIM process (see Figure 1). The main difference is the presence of the resin inlet in VARIM.

First, the titanium plates were pre-treated, applying wipe cleaning with MEK followed by grit blasting. After grit blasting, they were cleaned with compressed air and wipe cleaning with isopropanol. Finally, the application of a commercial sol-gel treatment, derived from a solution of organofunctional silane and zirconium alkoxide precursors, was carried out. 

The titanium alloy plates were assembled on the mould using anti-adherent tooling and then 5 layers of carbon fibre fabric were deposited over the metal. 

In the case of HLU manufacturing, the resin was deposited manually layer by layer and a consolidation roller was used in order to eliminate the possible air between the layers. After that, the vacuum bag and the vacuum inlet were assembled, and multi-material was subjected to consolidation with the vacuum bag. 

In the VARIM process, after depositing the carbon fibre layers, the resin inlet and the vacuum tube were assembled. The resin was degassed for 15 min at a temperature of 80 °C and then the resin infusion process was carried out.

The curing treatment applied was the same in both manufacturing processes, 2 h at 120 °C, 2 h at 180 °C, and 2 h at 200 °C. Then, multi-material samples were machined in the required dimensions for the different tests.

### 2.3. Characterisation

#### 2.3.1. Optical Microscopy and Image Analysis

The manufacturing process can influence the mechanical properties of multi-material joints because of heterogeneities into the interphase layer. As already mentioned, differences in thickness and possible porosity of the adherent layer, which acts as an adhesive between the titanium alloy and CFRP, could influence the fracture energy of the joints [21,22,27]. Besides this, different manufacturing processes such as HLU and VARIM, as well as the presence of CNT in epoxy resin, could result in differences in joint quality, which was studied by optical microscopy (MO).

Optical microscopy was carried out using the Olympus GX-71 microscope in cross sections of the multi-material joints. A specific image analysis software, ImageJ, was used for measuring the adhesive interface thickness and evaluating the possible porosity of the doped adhesive in order to compare the influence of manufacturing processes of CNT doped multi-material joints.

#### 2.3.2. Lap Shear Test

Single lap joints, 100 × 25 mm, were manufactured following the ASTM D5868-01 standard, with simple overlap geometry 25 × 25 mm, using substrates of 2-mm thickness for the Ti6Al4V alloy and approximately 2.5 mm for the multiscale-CFRP.

The tests were performed with a Zwick/Roell universal machine with a 100 kN load cell, applying a bridge displacement speed of 1.3 mm/min. Five joints of each combination were tested.

#### 2.3.3. Mode I Adhesive Fracture Energy (*G_IC_*) Test

Samples of 250 × 25 mm for double cantilever beam (DCB) tests were performed to determine the mode I fracture energy of the adhesive joints. At one end of the joint, a 40 mm long non-adhesive insert was placed, which would act as a crack initiator. A load was applied with a constant crosshead displacement rate of 1 mm/min in displacement control.

Corrected beam theory (CBT) analysis method was performed as described in ISO 25217:2009, and the following expression was used in order to calculate the *G_IC_* as a function of the crack length:(1)$$G_{IC}=\frac{3P\delta }{2B\left(a+\left|\Delta \right|\right)}\cdot \frac{F}{N}$$
where *P* is the load measured by the load-cell of the test machine; δ is the displacement of the crosshead of the test machine; *B* is the width of the specimen; *a* is the crack length; |Δ| is the crack length correction for a beam that is not perfectly built-in; *F* is large displacement correction; and *N* is the load-block correction.

#### 2.3.4. Structural Health Monitoring (SHM)

Monitoring of multi-material joints with multi-walled carbon nanotubes was achieved by measuring electrical resistance with a digital multichannel data acquisition system (Agilent 34972A-Keysight Technologies) by the two-point method. Monitoring of strain and damage was performed by locating copper electrodes fixed by silver paint in order to minimize the electrical contact resistance, on the surface of materials. Figure 2 shows the electrodes disposition in single lap joint and double cantilever beam specimen.

## 3. Results

### 3.1. Optical Microscopy and Image Analysis for Multi-Material Joint Quality

As already mentioned, the manufacturing process could influence the quality of multi-material joints. Two parameters related to the quality of the multi-material joints were evaluated: the porosity and the thickness of the resin layer on the interface. Table 1 shows micrographs of samples manufactured with different processes, HLU and VARIM, using neat epoxy resin and epoxy matrix doped with 0.1 wt% CNT, as a matrix. In all cases, the high quality of the samples, without porosity on the multiscale composite, and an excellent interphase metal-CFRP were confirmed. No differences were observed between the studied samples at this magnification level.

The thickness of the adhesive and the porosity of the polymer matrix were measured by using digital images of optical micrographs, whose results are shown in Figure 3. Comparing manufacturing processes, the multi-material joints manufactured by HLU present higher interface thickness than the processed ones by VARIM (Figure 3a). Notwithstanding the above, specimens manufactured by VARIM results in interface thickness values, in the range of 100 to 50 µm, much greater than those found by Streitferdt A. et al. [28], around 5 µm. Nevertheless, the thicknesses of all manufactured samples are within the minimum value range of those studied by other authors [19] in which they relate the influence of the adhesive thickness with the mechanical properties of the multi-material joints. On the one hand, the studies by da Silva et al. [19] and Gleich et al. [20] clearly indicate that the less the thickness of the adhesive layer, the greater the lap shear strength. Both have considered small thickness ranges in their studies: between 0.2 and 1 mm in the case of da Silva et al. and between 0.05 and 0.5 mm in the case of Gleich et al.

On the other hand, in the case of the studies found in which the fracture energy of the joint is related to the thickness of the adhesive layer, the conclusions differ slightly.

On the one hand, Mall et al. [21] have studied thicknesses of 0.1 and 0.25 mm, obtaining similar fracture energy results. On the other hand, when they increase to almost 0.5 mm in thickness, the result of the fracture energy clearly increases. In the case of the study carried out by Carlberger and Stigh [22], it was found that the fracture resistance increases monotonically as the thickness increases from 0.1 to 1 mm, reaching a maximum when a range between 1–1.6 mm is reached. In these cases, the previous studies are in higher thickness ranges, the maximum reached in this work being the minimum of the range studied.

The interface thickness on samples manufactured by HLU is slightly affected by the presence of CNT, mainly due to the difficulty of applying large layers of resin with higher viscosities.

On the other hand, a large decrease in thickness is observed in the VARIM process with CNT doped resin. This effect is caused by two factors: the increase of the resin viscosity, which hinders its flow during the infusion process and the filtering effect, which also causes a resistance to the flow of the resin, inducing a decrease in the absolute pressure within the bagged system, resulting in greater compaction and a much lower layer thickness at the interface.

The porosity was calculated by the average percentage of porosity of five sections for each type of joint in which the total area and the areas of the pores present in the joint were measured. As can be seen in Figure 3b, the porosity is somewhat higher in the case of the HLU manufacturing process. In spite of the deaerator roller being used to eliminate air bubbles, pressure compaction was applied by vacuum bag and the resin was previously degassed and preheated, this is not enough to completely avoid the presence of porosity, reaching 3% of porosity. As was expected, the porosity of the materials manufactured by VARIM was lower. The use of CNT doped resin induced a slight increment of the porosity of the matrix due to the increase of the non-cured resin viscosity, which hindered the resin injection. Moreover, the possible filtering effect of carbon nanotubes in the thickness direction can also induce an increase in the matrix porosity. This justification is also confirmed because the porosity does not depend on CNT addition for the multi-material joints manufactured with HLU, meaning that the porosity measured by VARIM on doped composites must be associated with the resin infusion process.

The increase of porosity on VARIM manufactured samples by CNT addition may be due to the change in the viscosity, as reflected in previous studies [29], of the inlet preheated resin as the temperature decreases with time. This change in viscosity affects the impregnation of the carbon woven and could result in non-impregnated areas or air bubbles, especially in areas further away from the resin inlet. The filtering effect would be another effect to consider and that could influence the presence of porosity, due to the formation of CNT clusters that obstruct the correct flow of the resin and a homogeneous impregnation.

### 3.2. Mechanical Properties

#### 3.2.1. Influence of Manufacturing Process on Lap Shear Stress

Figure 4 shows the average values of lap shear strength (LSS) of the Ti6Al4V-CFRP joints using neat epoxy resin and the CNT doped one, manufactured by HLU and VARIM. No great differences were observed by this mechanical test. It is important to state that this test only evaluates the mechanical joint strength without influencing the possible differences in deformation ability and toughness of the adhesive. The optical images of the fractures after the LSS test (Figure 4) show, in all cases, a similar failure mode with adhesive failure and the presence of microcohesivities.

There is a slight improvement in the case of joints manufactured by the VARIM process. Specifically, joints with 0.1% CNT manufactured using the VARIM resulted in an 8% higher lap shear strength than those manufactured by HLU. This may be mainly due to the quality of the joint since the VARIM processed joints have a lower porosity (Figure 3b). Besides this, there was a significant difference in the interface thickness (Figure 3a) comparing with HLU. VARIM manufactured joints showed a lower value for interface thickness, and this behaviour coincides with the previous studies found on the influence of the thickness of the adhesive layer in SLJ [19].

On the other hand, using both manufacturing processes, there is a slight detriment to the lap shear strength when CNT doped resins are used. This effect could be addressed as the impediment by the CNTs to the chemical bond between the sol gel and the resin, because CNTs occupy part of the direct bonding surface between epoxy matrix and the sol-gel treated Ti6Al4V substrate. As shown by Rachmadini et al. in their review [30], several authors have shown a strengthening effect of CNTs in their studies. However, these effects are seen in mode I or mode II fracture toughness tests. On the other hand, most studies do not consider the differences in lap shear strength due to different thicknesses when CNT is dispersed, as reported by Da Silva et al. in their studies based on neat resin (without CNT doped resin) [19].

This decrease in lap shear due to the presence of nanotubes is less in the case of the VARIM processed joints than the samples manufactured by HLU. This may be due to several factors such as the cumulative effect of higher porosity and the impediment of CNTs to chemical bonding at the interface in the case of the HLU process.

#### 3.2.2. Influence of Manufacturing Process on Fracture Energy

Figure 5 shows the fracture energy in mode I for neat and reinforced with 0.1% CNT multi-material DCB joints manufactured by HLU and VARIM processes. CNT addition on the epoxy matrix clearly improves the resistance to fracture of multi-material joints. Despite that, in the case of using neat epoxy, the results are similar, in the case of using the resin doped with carbon nanotubes, a significant difference was found between HLU and VARIM processes. A higher value of fracture energy was obtained for joints manufactured by HLU, with an increment of 31.5% in the case of joints with 0.1% CNT manufactured by VARIM. This is due to the fact that, as can be seen in Figure 3a, the interface thickness is greater in HLU manufactured joints than those manufactured by VARIM. As previously mentioned, several authors [21,22,27] have concluded in their studies that an increase in the thickness of the adhesive layer leads to an improvement in mode I fracture energy. On the other hand, the effect of CNTs causes a mechanical anchoring and stiffening effect which has a great influence on mode I fracture tests. This effect is not visible in the case of LSS tests, where the mechanical anchoring of CNT is not enough to influence on shear failure mechanism. Due to the filtering effect in thickness direction [30], in the case of the VARIM process, the thinner interface implies a smaller amount of CNT on this layer, which results in a detrimental effect.

This effect can also be appreciated in the optical images of the samples after the mode I fracture test. As can be seen in Figure 5, there is an appreciable difference in cohesive failure between the different manufacturing methods. The percentage or area of cohesive breakage was calculated by filtering and treating the image, obtaining differences of around 10% between the joints manufactured by the HLU process (60.80% of cohesive failure) and those manufactured by the VARIM process (50.45% of cohesive failure).

### 3.3. Influence on Structural Health Monitoring (SHM)

#### 3.3.1. Influence of Manufacturing Process on SHM of Lap Shear Stress Test

The structural health monitoring of the LSS tests for multi-material joints manufactured by the different manufacturing processes under study can be seen in Figure 6.

No significant differences were found regarding the structural monitoring of multi-material joints under the lap shear stress test. Although, the sensitivity expressed by the gauge factor (Figure 6c) is higher in the case of joints manufactured by the VARIM process. This variation is quite remarkable, reaching up to 56%. The main reason for this greater sensitivity in joints manufactured by the VARIM process is due to their lower insulating layer thickness (Figure 3a). In this way, the tunnelling effect, due to a smaller insulating layer, and the direct contact between CNT, titanium alloy, and carbon fabric, are promoted. HLU joints have a higher interface, increasing the distance between the main conductor components, metallic alloy, and carbon fabric, increasing the electrical resistance, and decreasing the sensor sensibility.

On the other hand, the lower gauge factor value obtained in the case of joints manufactured by the HLU process could be attributed to the higher porosity of this type of joints (Figure 3b) and the relatively small test area. In addition, in the case of joints manufactured by VARIM, a lower presence of CNT in the interface, which could be caused by the filtering effect, leads to the breakage of the low number of conductive paths during the shear test, resulting in greater variability of the electrical signal. It is well known that the less CNT amount, above the electrical percolation threshold, enhances the SHM sensibility [31].

#### 3.3.2. Influence of Manufacturing Process on SHM of Fracture Energy-Mode I Tests

Figure 7 shows the mechanical and SHM curves for fracture energy-Mode I tests. In this case, several sensor channels are positioned (Figure 2c). The 3 channels (# 1 blue, # 2 green, and # 3 pink) have been monitored and the specific positions of the connectors are indicated by the dashed lines in Figure 7a,c. 

The tendency of the electrical sensors has been previously observed in other studies [32], where the compressive load in thickness direction results in a drop in electrical resistance followed by an increase in electrical resistance as the crack spreads.

This behaviour is captured by the channels as the crack approaches each of them and has been indicated by a dashed box and the number of the corresponding channel in Figure 7b,d. On the other hand, it can be seen that channel # 1 presents greater variations when the crack propagates in its vicinity; followed by channel # 2. Channel # 3 is far enough away from the beginning of the crack to progress slowly in resistance variation and present the most abrupt change when the crack approaches connector # 3.

This behaviour is captured in both types of joints, either manufactured by HLU or by VARIM. It could then be stated that both are sensitive to crack propagation.

On the other hand, trend changes due to compressive stress at the crack edge followed by upshift due to propagation, appear slightly higher for VARIM-manufactured joints (Figure 7d). This may be due to a lower presence of CNT and the lower thickness in the interface.

However, changes in normalized electrical resistance as a function of fracture energy are more dramatic for HLU joints (Figure 7a,b). This may be related to its higher fracture energy and the more unstable progress of the crack, resulting in jumps in the mechanical curve that are faithfully represented by the electrical curves, especially in the first channel. In the case of the follow-up of the fracture resistance of the joint for the case of VARIM joints, this progress seems to be not so coincident, especially when it exceeds the # 2 connector. This may be due to greater heterogeneities in the joint caused by variations in the arrangement of the CNTs throughout the joint.

It could be concluded that structural health monitoring (SHM) of joints manufactured using the HLU process result in slightly better fracture stress monitoring. However, in the case of joints manufactured by VARIM, they are more sensitive to crack propagation and stresses caused by the crack front.

Table 2 collects the fracture energy measured and the electrical parameters registered. Joints manufactured by the HLU process have shown greater electrical resistance than VARIM joints. This result may be explained by the fact that, despite being able to present a greater amount of CNT in the interface, the amount of resin and therefore of the insulating layer is also greater (see Figure 3a).

In contrast, in the case of joints manufactured using the VARIM process, the lower interface thickness (Figure 3a) results in a smaller insulating layer and greater proximity between the Ti6Al4V and the carbon weave. This enhances electrical contact between carbon fiber-CNT- Ti6Al4V, making the electrical resistance much lower.

When the average variation in normalized resistance is evaluated, slightly higher value was obtained in the case of joints manufactured using the HLU process. However, the mechanical resistance values have also been higher, which results in similar sensitivity in both cases: joints manufactured by HLU and VARIM processes.

## 4. Conclusions

This research has analysed the influence of the multi-material joint manufacturing process on the quality, mechanical properties, and electrical sensitivity of structural health monitoring (SHM) by CNT addition of the epoxy matrix. The processes analysed were HLU assisted with vacuum bag and VARIM to manufacture co-cured joints constituted by titanium alloy, Ti6Al4V, surface treated with specific sol-gel and multiscale CFRP joints whose matrix is doped with 0.1 wt%.

When the variations in the mechanical behaviour of the joints have been analysed, it can be concluded that in LSS the differences are not significant. This is mainly associated to the presence of higher porosity in the HLU manufacturing process, which should decrease the strength, but it is compensated by the smaller interface adhesive thickness, which gives rise to a greater lap shear strength.

This lower thickness is also responsible for the variations seen in the sensitivity of the joints during structural health monitoring. The sensitivity of VARIM manufactured joints is higher than that of HLU process joints. This is associated with two factors. On the one hand, the lower thickness of the interface gives rise to greater proximity between layers of conductive substrates (carbon woven and titanium alloy), facilitating the tunnelling effect. On the other hand, the possible lower presence of CNT in the interface, due to the filtering effect, causes the conductive paths to break earlier, which are found in less quantity, giving rise to a greater variation in resistance and therefore greater sensitivity.

In the case of joints subjected to the fracture test in mode I, the values obtained for GIC are clearly higher in the case of joints manufactured using the HLU process. This is due to the marked interface thickness increase, which has a great influence on the fracture resistance of the joint. In addition, the presence of CNT plays an important role, since as observed in the case of joints made with neat epoxy, this difference is not so pronounced, as is the thickness of the interface, which does not vary significantly in the case of comparing both processes with neat epoxy joints.

Regarding sensitivity during SHM of mode I tests, no large variations have been detected between the two processes. Even though VARIM process joints have a lower initial electrical resistance, the quality of the joint manufactured by HLU and its great mechanical behaviour against fracture in mode I, results in changes in electrical resistance captured just as well as joints with lower electrical resistance.

## Figures and Tables

**Figure 1 polymers-13-02488-f001:**
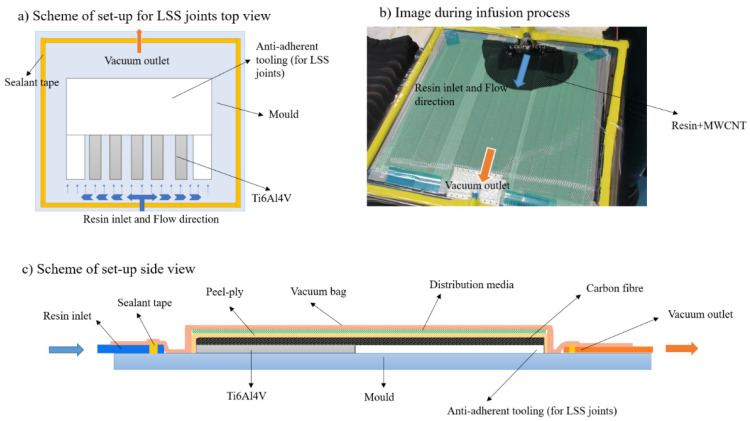
(**a**,**c**) Scheme of VARIM process and (**b**) image with resin flow during VARIM process

**Figure 2 polymers-13-02488-f002:**
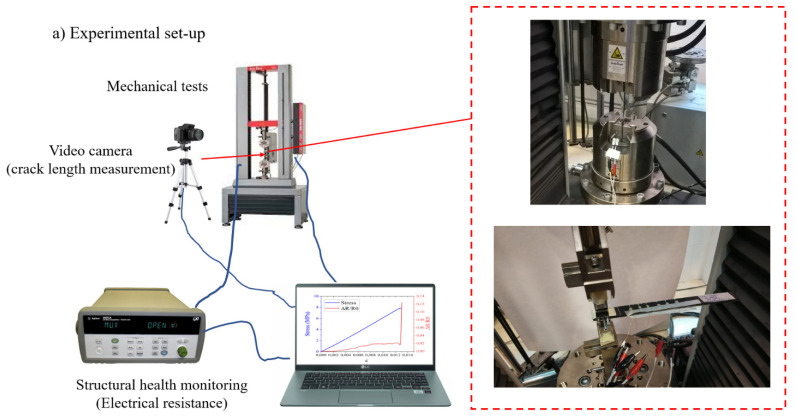
(**a**) Experimental set-up and scheme of electrode configuration for electrical monitoring during (**b**) single lap shear test and (**c**) Mode I adhesive fracture energy (GIc) test with DCB probes.

**Figure 3 polymers-13-02488-f003:**
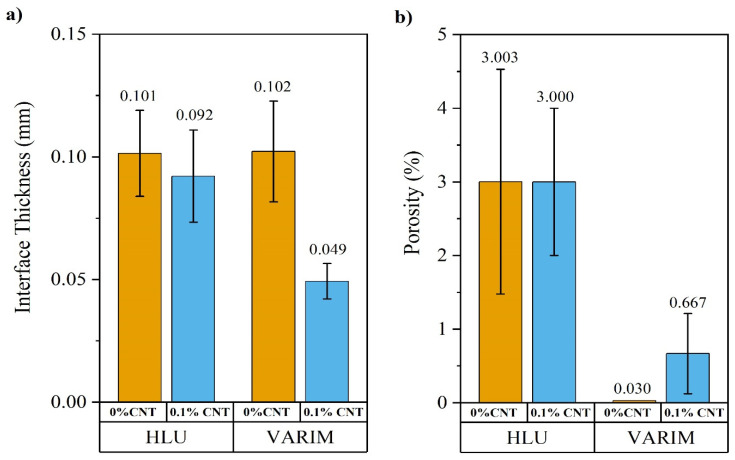
Quality of joints manufactured by HLU and VARIM: Interface thickness (**a**) and percentage of porosity (**b**).

**Figure 4 polymers-13-02488-f004:**
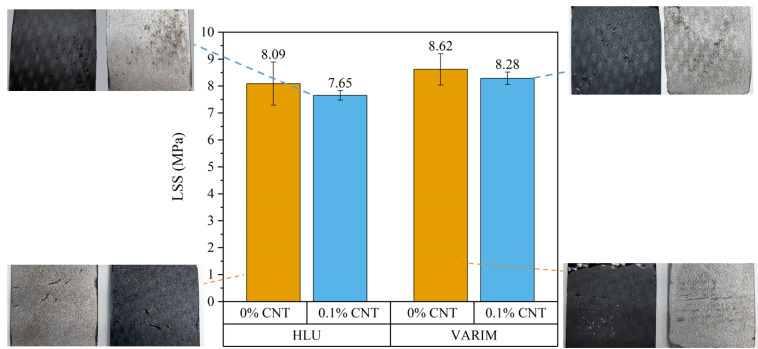
Lap shear strength for neat and reinforced with 0.1% CNT multi-material joints manufactured by HLU and VARIM.

**Figure 5 polymers-13-02488-f005:**
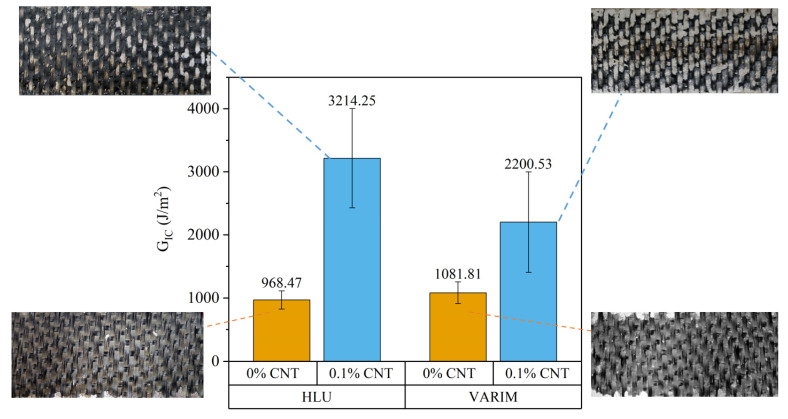
Fracture energy in mode I (GIC) for neat and reinforced with 0.1% CNT multi-material joints manufactured by HLU and VARIM processes.

**Figure 6 polymers-13-02488-f006:**
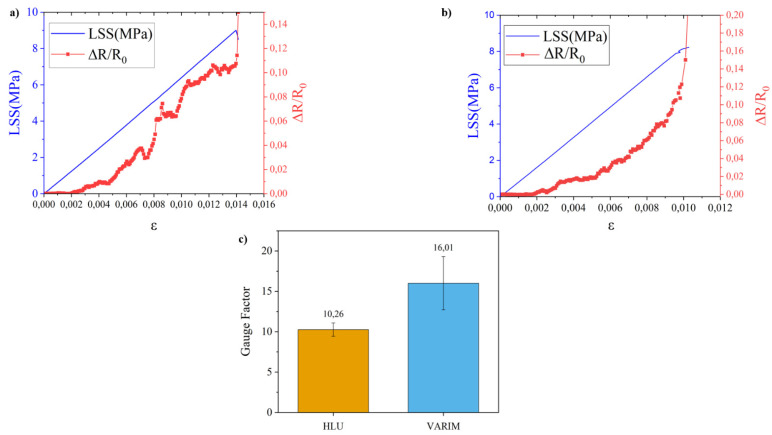
Mechanical and electrical SHM curves obtained during LSS tests for HLU (**a**) and VARIM (**b**) manufactured multi-material joints and average gauge factor (**c**).

**Figure 7 polymers-13-02488-f007:**
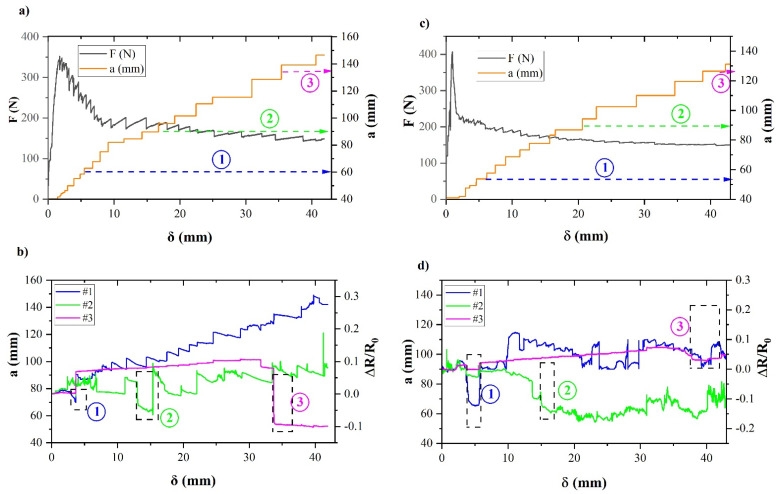
Mechanical (**a**,**c**) and electrical resistance (SHM) curves (**b**,**d**) for fracture energy test of multi-material joint manufactured by HLU (**a**,**b**) and VARIM (**c**,**d**) manufacturing processes, using a resin reinforced with 0.1% CNT and positioning of crack opening during test (**e**).

**Table 1 polymers-13-02488-t001:** Optical microscopies (100×) of multi-material joints manufactured with different manufacturing processes: HLU and VARIM with both neat epoxy and 0.1% CNT doped resin (image scale bar shows 200 µm).

	Neat Resin	CNT Doped Resin
HLU	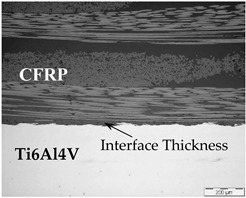	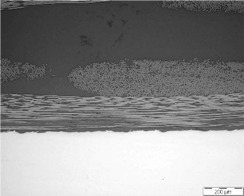
VARIM	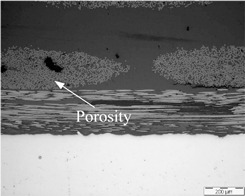	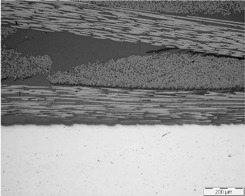

**Table 2 polymers-13-02488-t002:** Average values for initial resistance (R_0_), normalised resistance (ΔR/R_0_), and fracture energy in mode I (G_IC_) for DCB multi-material joints reinforced with 0.1% CNT and manufactured by HLU and VARIM processes.

	HLU	VARIM
R_0_ (Ω)	12.51 ±2.1	1.19 ± 0.35
G_IC_	3214 ± 788	2200 ± 796
ΔR/R_0_	1.20 ± 0.26	0.97 ± 0.35

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
