# Peer review of "Influence of Manufacturing Process in Structural Health Monitoring and Mechanical Behaviour of CNT Reinforced CFRP and Ti6Al4V Multi-Material Joints"

_polymers, 2021, doi:10.3390/polym13152488_

Round 1

Reviewer 1 Report

The paper "Influence of manufacturing process in structural health monitoring and mechanical behaviour of CNT reinforced CFRP and 3 Ti6Al4V multi-material joints" analyze the influence of manufacturing processes on the morphological quality and mechanical behaviour of sol-gel treated Ti6Al4V and carbon fibre reinforced composites (CFRP). In the opinion of this referee it can be published after major revision. Please condire the following issues to be addressed.

Generally, the quality of the figures has to be improved. 

The adhesion of the different materials could depend on the mechanism involved in the dispersion process, which has to be better explained. Please consider the following (https://doi.org/10.1016/S0032-5910(97)03210-5; https://doi.org/10.1002/aelm.201600126).

Figure 7 is not clear at all, it would be better to split each figure into two figures.

Did the authors made some optical/electronic  image of the samples after mechanical tests? They could help in explain the behavior of the materials. 

Author Response

  1. We have improved all the figures as follows:
    1. Figure 1: The quality of the figure (scheme) has been improved and an image during the infusion process for VARIM manufacturing has been included for a better understanding.
    2. Figure 2: An experimental set-up scheme was included and channels for DCB-Mode I test have been identify with colors (corresponding with signals on electrical curves).
    3. Figure 3: the data on the bar plots have been implemented and quality of the image has been improved.
    4. Figure 4 and 5: optical images of the samples after the mechanical test and data on the bars plot have been included.
    5. Figure 7: as suggested by reviewer 1, each figure was split in two figures. One for the mechanical curve and the second one for electrical signals. An image and numeration in order to identify the position of the crack opening during mode I test have been included.

  1. The authors have included a description of the dispersion process of CNTs in the resin, in accordance with the proposal of the reviewer 1, citing some previous works where the development and study of the calendering process has been carried out, as well as the study of the influence of the filtering effect on the material properties. (Lines 96-102)        

CNTs were dispersed into the resin by calendering method. Calendering is a mechanical dispersion method, where the resin with CNT passes a certain number of times between the rollers of the calender. This method was based on the use of a three-roll calender machine (Exakt 120S). In each step, the distance between rollers is reduced, so that gradually the agglomerates of CNT are dissolved. The gap size varied in steps (in microns): (1) 120 - 40, (2) 60 - 20, (3) 45 - 15, and (4) 15 - 5. The last step was repeated four consecutive times. The speed used was 250 rpm.

These parameters and the quality of the dispersion have been studied by the authors in previous works which has already been described in previous works [23–25]. 

[23] Jiménez-Suárez A, Campo M, Sánchez M, Romón C and Ureña A 2012 Influence of the functionalization of carbon nanotubes on calendering dispersion effectiveness in a low viscosity resin for VARIM processes Compos. Part B Eng. 43 3482–90

[24] Prolongo S G, Rosario G D and Ureña A Coupled thermal-electrical analysis of carbon nanotube/epoxy composites Polym. Eng. Sci. 54 1976–82

[25] Jiménez-Suárez A, Campo M, Gaztelumendi I, Markaide N, Sánchez M and Ureña A 2013 The influence of mechanical dispersion of MWCNT in epoxy matrix by calendering method: Batch method versus time controlled Compos. Part B Eng. 48 88–94

  1. The authors totally agree with the comment of reviewer 1 about the convenience of the images of the samples after tests. Images (macro) were included in figures 4 and 5. Specially in mode I fracture tests (Figure 5), the differences between one manufacturing process and another are marked, evidencing the detrimental effect of the filtering effect in the case of VARIM manufacturing (Lines 304-309)

This effect can also be appreciated in the optical images of the samples after the mode I fracture test. As can be seen in Figure 5, there is an appreciable difference in cohesive failure between the different manufacturing methods. The percentage or area of cohesive breakage has been calculated by filtering and treating the image, obtaining differences of around 10% between the joints manufactured by the HLU process (60.80% of cohesive failure) and those manufactured by the VARIM process (50.45% of cohesive failure).

Reviewer 2 Report

The paper presents an experimental study of the effect of manufacturing process (hand layup or VARIM) and the use of carbon nano-tubes, on the mechanical and electrical properties of co-cured joints of carbon fibre and Titanium. A variety of procedures have been used and the results are clearly described. Comments:
line 35 – should be widely
line 93 – should that be Huntsman?
line 181 -should be No
Table 1 – the images require a scale, also it would be beneficial to indicate what the relevant parts are for clarity
line 183 – authors use terms like thickness of adhesive, interlayer thickness and interface thickness – please define and be consistent with your terminology
line 217 – explain how porosity was calculated
Figures – the data in the bar plots are not provided in the paper – please add the actual numbers, either to the plots or in a table
line 255 – most authors find adding CNTs has a strengthening effect, please explain
line 382 – delete greater

Author Response

Reviewer 2:

  1. Spell check indicated in lines 35, 93, 181 and 382 have been corrected.
  2. The authors have indicated the relevant parts in the figures in Table 1 and the scale has been included.
  3. The terminology has been combined, selecting “interface thickness” as the appropriate term to define the thickness of resin with CNT present between CFRP and Ti6Al4V.
  4. Explanation for the calculation of porosity has been added (Line 225)

The porosity has been calculated by the average percentage of porosity of five sections for each type of joint in which the total area and the area of the pores present in the joint have been measured.

  1. Reviewer 2 indicates that most authors find that adding CNTs leads to a strengthening effect, which in our work only seems to be reflected in case of mode I fracture energy.

An explanation based on different studies have been added: Most of the literature presents an adhesive application in case of dissimilar joints, with a greater thickness and viscosity than in the case of manufacture by VARIM using a thin layer of resin. In case of a “traditional adhesive” the reinforcing effect of CNTs is more notable due to the possibility of stiffening the adhesive and a greater distribution of the crack propagating effects. Regarding the literature using the VARIM method in joints, most of the satisfactory results are referred to fracture tests in mode I or mode II and the improvement in in-plane fiber-dominated properties is very modest, as reflected in the review:

[30] Rachmadini Y, Tan V B C and Tay T E 2010 Enhancement of Mechanical Properties of Composites through Incorporation of CNT in VARTM - A Review J. Reinf. Plast. Compos. 29 2782–807

(Line 273) As shown by Rachmadini et al. in their review [30], several authors have shown a strengthening effect of CNTs in their studies. However, these effects are seen in mode I or mode II fracture toughness tests. On the other hand, most studies do not consider the differences in lap shear strength due to different thicknesses when CNT is dispersed, as reported by Da Silva et al. in their studies based on neat resin (without CNT doped resin) [19].

Round 2

Reviewer 1 Report

The authors improved the manuscript a lot, however, the quality of the figures is still poor. I suggest to enhance the quality of the figures.

Author Response

The quality of images (macro- and micrographs) has been increased. Now, they have more resolution. Thank you for the suggestion.